# Postural Adjustments in HTLV-1 Infected Patients during a Self-Initiated Perturbation

**DOI:** 10.3390/v14112389

**Published:** 2022-10-28

**Authors:** Gizele Cristina Da Silva Almeida, Hélio Resque Azevedo, Kelly Helorany Alves Costa, Alex Tadeu Viana da Cruz Júnior, Daniela Rosa Garcez, Givago da Silva Souza, Bianca Callegari

**Affiliations:** 1Laboratory of Human Motricity Studies, Institute of Health Science, Federal University of Pará, Belém 66050-160, Brazil; 2Tropical Medicine Institute, Federal University of Pará, Belém 66050-160, Brazil; 3University Hospital Bettina Ferro de Souza, Federal University of Pará, Belém 66075-110, Brazil; 4Neuroscience and Cell Biology Graduate Program (PPGNBC), Federal University of Pará, Belém 66075-110, Brazil

**Keywords:** Human T Lymphotropic Virus Type 1, anticipatory postural adjustment, postural control

## Abstract

Background: Human T-cell lymphotropic virus type 1 (HTLV-1) infection can be associated with tropical spastic paraparesis (TSP/HAM), which causes neurological myelopathy and sensory and muscle tone alterations, leading to gait and balance impairments. Once trunk perturbation is predicted, the motor control system uses anticipatory and compensatory mechanisms to maintain balance by recruiting postural muscles and displacement of the body’s center of mass. Methods: Twenty-six participants (control or infected) had lower limb muscle onset and center of pressure (COP) displacements assessed prior to perturbation and throughout the entire movement. Results: Semitendinosus (ST) showed delayed onset in the infected group compared to the control group. The percentage of trials with detectable anticipatory postural adjustment was also lower in infected groups in the tibialis anterior and ST. In addition, COP displacement in the infected group was delayed, had a smaller amplitude, and took longer to reach the maximum displacement. Conclusions: HTLV-1 infected patients have less efficient anticipatory adjustments and greater difficulty recovering their postural control during the compensatory phase. Clinical assessment of this population should consider postural stability during rehabilitation programs.

## 1. Introduction

Human T-cell lymphotropic virus type 1 (HTLV-1) is a delta retrovirus associated with several pathologies, including adult T-cell leukemia/lymphoma and tropical spastic paraparesis/HTLV-1–associated myelopathy (TSP/HAM), among other clinical conditions [1]. It is estimated that 15 to 20 million people worldwide are infected with HTLV-1, which is endemic to southern Japan, the Caribbean, Central Africa, Central and South America, Melanesia, and the aboriginal population of Australia [2]. Transmission can occur through sexual contact, blood materials, and vertical contagion [1]. This infection affects the structure and function of several body systems, including the central nervous system (CNS) [3]. TSP/HAM is a myelopathy associated with HTLV-1 infection that causes a neurological disorder ,due to demyelination and axonal destruction of the spinal cord [4,5]. Motor impairment is a focus of many investigations reporting functional deficits in patients with TSP/HAM. 

Body balance is defined as the ability to keep the center of mass (COM) stable within the available basis of support, and is a functional skill that depends on motor and sensory components [6]. The motor control system uses anticipatory (APA) and compensatory (CPA) postural adjustment strategies to maintain and restore balance. APAs are performed through the activation of postural muscles prior to a predicted disturbance in a feedforward mechanism to reduce possible imbalances [7]. When performing a rapid arm movement, body perturbation occurs, which is predicted by the CNS and adjusted in an anticipatory manner through a feedforward mechanism [7,8,9,10]. CPAs are generated after the disturbance by a sensory feedback mechanism, and involve contraction of muscles to reduce the generated imbalance [11,12]. 

APAs have been studied extensively in patients with neurological disorders. A lower magnitude of muscle activation, in addition to a delay in the onset of activation, has been demonstrated in post-stroke and multiple sclerosis patients [13,14,15,16]. The literature excludes patients with HTLV infection, and although APA has never been investigated, there is evidence that the balance in these individuals is altered. Vasconcelos et al. demonstrated that these patients presented higher center of pressure (COP) displacement, COP area, and COP velocity in a stabilometric analysis [17]. These results led us to hypothesize that anticipatory and compensatory adjustments are compromised in this population.

Therefore, it is crucial to investigate anticipatory postural adjustments in individuals infected with HTLV-1, to address postural control deficits in these patients. Therefore, we aimed to measure and compare postural adjustments in healthy adults and HTLV-1 infected patients, during a pointing task paradigm in a standing posture.

We hypothesized that HTLV-1 infection triggers changes in postural control mechanisms in infected patients. Specifically, we hypothesized that the infected individuals would present with delayed and lower muscle activity, accompanied by delayed and increased COP displacements during the APA phase. We further hypothesized that these APA responses would reflect compensatory adjustments, owing to a higher COP displacement during the CPA period. To investigate this, we used a self-perturbation paradigm in which the subjects were requested to perform a pointing task while postural muscles and COP were measured.

## 2. Materials and Methods

This cross-sectional observational study was conducted at the Laboratory of Human Motricity Studies (LEMOH) of the Federal University of Pará. This study was approved by the Ethics Committee of the Federal University of Pará (protocol #3.621.532) and conforms to the Observational Studies in Epidemiology (STROBE) Statement. Written informed consent was obtained from all participants before the start of the study. 

### 2.1. Participants

Twenty-six individuals (13 healthy and 13 infected with HTLV-1), aged between 18 and 59 years, participated in the study and were divided into two groups: control and infected. The groups were matched for age, body mass, and sex. The HTLV-1 infected subjects were recruited from the Tropical Medicine Institute of the Federal University of Pará, Brazil. The patients were diagnosed by a doctor, based on the World Health Organization (WHO) criteria. An infectologist collected data on the patients’ clinical histories, neurological evaluation results, and seropositivity results, using enzyme-linked immunosorbent assay (ELISA) (Cambridge Biotech, Worcester, MA, USA), western blot analysis (HTLV blot 2.4, Genelab, Singapore), polymerase chain reaction (PCR), or a combination of these examinations, for HTLV-1 infection. Patients with more than three neurological symptoms were classified as having TSP/HAM. We excluded patients who were unable to remain in a standing position without assistance; those with neurological disorders with diagnoses other than HTLV-1 infection, such as diabetes, rheumatic diseases, and peripheral vestibular syndrome; HIV-coinfected patients; pregnant women; individuals with other diseases that may affect balance and motor control; and those with cognitive impairment [18]. All individuals in the infected group were characterized according to the time of infection and presence of neurological symptoms (Table 1).

### 2.2. Experimental Setup for Postural Adjustments Assessment

The participants stood on a force platform and performed the task of quickly pointing to the light emitting diode (LED) that lit up, as shown in Figure 1, in 10 attempts. For this, it was mandatory to remain concentrated and staring at the LED positioned on a horizontal bar 2 m away from the subject, at a height of 2.5 m. The movement was performed as fast as possible, always keeping the elbow in extension and maintaining the pointing position for 2 s before returning to the starting position.

### 2.3. Data Collection Procedures

A surface electromyograph (Emgsys 30306^®^; EMG System do Brasil, São José dos Campos, Brazil), kinematics system with four cameras (Simi Motion), and a force platform (EMG System do Brasil) was used to extract the features of movement trajectory, muscle activity, and COP displacement during the task. All signals were synchronized over time.

Electromyographic acquisition was performed at a frequency of 2000 Hz, and the skin of the participants was prepared for the application of Ag/AgCl electrodes (Med-Trace 200; Covidien Kendall, Dublin, Ireland) by shaving with disposable materials and cleaning with 70% alcohol. Disposable self-adhesive electrodes were fixed on the muscle bellies of the deltoid (DEL), semitendinosus (ST), soleus (SOL), rectus femoris (RF), and tibialis anterior (TA), in the hemibody ipsilateral to the upper limb that performed the movement, keeping a distance of 2 cm between the electrodes, following the recommendations of the Surface Electromyography for the Non-Invasive Assessment of Muscles (SENIAM) project [19]. To record participants’ movement speed, a three-dimensional motion analysis system (Simi Motion, Simi, Germany) was used, with four cameras at a sampling frequency of 120 Hz. The reflective marker was positioned at the distal end of the index finger of the limb that was used to perform the pointing movement. A force platform (BIOMEC400, EMG System do Brasil, Ltd.a., São Paulo Brazil) was used, with load sensors distributed in a 50 cm² area and connected to a computer using Biomec software (EMG System do Brasil, Ltd.a., São Paulo, Brazil). COP displacements were collected while performing the task, with an acquisition frequency of 100 Hz. The individuals remained standing barefoot on the platform, with their feet apart at a distance proportional to the distance between the shoulders and arms along the body (initial position).

### 2.4. Signal Processing

Surface electromyographic (EMG) data were recorded raw and subsequently filtered with 20 and 400 Hz bandpass, full-wave rectified, and bidirectionally offline filtered, using a 6 Hz second-order lowpass, zero-lag Butterworth filter. This filtering allowed for smoothing of the curve to identify the latency of muscle activity (activation or inhibition) and was determined by visual inspection performed by two blinded examiners. The beginning of movement was called T0 and corresponded to the beginning of deltoid muscle activation. After identifying the T0 of each attempt, the start time of the activation of other muscles was calculated, having T0 (DEL) as the reference, as a variable called latency. Latency was defined as the moment when muscle activity reached values corresponding to two standard deviations (2 SD) above (activation) or below (deactivation) the mean baseline value (measured from −500 to −450 ms before T0), and maintained this activation for at least 50 ms [20]. All trials in which muscle latency was observed up to 250 ms before T0 were considered valid APA. Finally, the percentage (%) of APA attempts was calculated for all muscles and individuals. Kinematic data from the index finger and force platform data were fed to two EMG channels to allow data synchronization and offline analysis using MatLab programs (MathWorks, Natick, MA, USA). The participants’ peak indicator velocity average was taken from raw x-, y-, and z-coordinate data generated from video analysis and then filtered, using a bidirectional low-pass Butterworth filter of the second order of 10 Hz. Figure 2 shows the detection of T0 (DEL) after the LED stimulus and the speed of the index finger.

COP displacements in the anteroposterior direction were recorded and calculated, based on previous studies [21]. The baseline record used for the calculation ranged from −500 to −400 ms in relation to T0, after which it was possible to calculate the variables of anticipatory (COP_onset_ and COP_APA_) and compensatory (COP_disp_ and COP_timetopeak_) characteristics (Figure 3).

(1)The start of the COP shift before time T0 was measured as the time when the COP shift was less than the mean of its baseline value plus 2 SD (COP_onset_) [22].(2)The anteroposterior displacement of the COP at time T0 (measured from the baseline amplitude) is known as the amplitude of the displacement of the COP at T0 (COP_APA_).(3)Peak displacement is measured as the maximum displacement after the moment T0 (COP_disp_)(4)Time taken to reach the peak of the maximum displacement is recorded as COP_timetopeak_.

### 2.5. Statistics 

GraphPad Prism 9 (San Diego, CA, USA) and BioEstat 5.3 (Belém, Pará, Brazil) were used to perform the statistical analysis in this study. COP_onset_ variable was calculated in a trial including five subjects in each group and revealed −0.764 s ± 0.20 (control group) and −0.512 s ± 0.22 (infected group). Adopting a power test rate of 80% and alpha 0.05, the sample size was established at 11 individuals in each group. Comparisons were performed between the control and infected group. The Shapiro–Wilk test was employed to verify the normality of the data, and the t-test was used for normally distributed data (COP variables), while the Mann–Whitney test was used for non-parametric data (electromyographic variables). For all these statistical treatments, the adopted significance level was set at *p*-value < 0.05.

## 3. Results

The groups showed a predominance of women, and no significant differences in age, weight, and height were present between the groups (*p* > 0.05). The finger peak velocity of movement also showed no difference between the groups (Table 2).

### 3.1. COP Variables

The COP moved in a posterior direction before the beginning of the pointing movement and gradually returned to the 0 value point after its maximum posterior displacement. Figure 4 shows the COP behavior of a control and an infected individual during one trial. Note that the control group participants had an earlier COP_onset_, a greater COP_APA_ amplitude, and took less time to reach maximum displacement after T0 (shortest COP_timetopeak_). No differences were observed between the COP_disp_ of the two groups.

Figure 5 presents the COP_onset_, COP_timetopeak_, COP_APA_, and COP_disp_ comparisons between the groups. The infected group showed delayed COP_onset_ (control: −0.2272 ± 0.07446 s; infected: −0.1400 ± 0.05944 s, t = 3.298; *p* = 0.0030*) and lower COP_APA_ (control: 0.3552 ± 0.1352 cm; infected: 0.1939 ± 0.1232 cm, t = 3.180; *p* = 0.0040*) than the control group. Although both groups achieved a similar COP_disp_ after the perturbation (0.8704 ± 0.2479 cm; infected: 0.8154 ± 0.4506 cm, t = 0.3856; *p* = 0.7032), the infected group needed more time to reach it, presenting a higher COP_timetopeak_ (control: 0.1440 ± 0.05736 s; infected: 0.2374 ± 0.07604 s, t = 3.536; *p* = 0.0017*). COP_onset_, COP_timetopeak_, and COP_APA_ showed worse performance in the infected group (Figure 5).

### 3.2. Muscle Latency

Figure 6 depicts the latency of each muscle in a single trial of a control and an infected participant. Note the anticipation of the control participant, represented in black (compared with the infected participant, represented in gray), in the ST muscle. No differences were observed between the groups in terms of the latency of the other muscles.

Figure 7 illustrates the latency of each muscle of the infected group compared with that of the control group (Figure 7). The latency of APA activity was as follows: ST (control: −0.1010 ± 0.06062 s; infected: −0.06154 ± 0.07915 s, U = 43.50, *p* = 0.0340*); RF (control: −0.09651 ± 0.06525 s; infected: −0.1017 ± 0.07614 s, U = 74.50, *p* = 0.8611); TA (control: −0.08399 ± 0.05914 s; infected: −0.1208 ± 0.1233 s, U = 77.50, *p* = 0.7332); SOL (control: −0.1080 ± 0.06268 s; infected: −0.1031 ± 0.04939 s, U = 75.50, *p* = 0.6579).

The percentage of trials with detectable APA was lower in the infected groups, except for the RF muscle (Table 3).

## 4. Discussion

This aim of this study was to investigate the behavior of APA and CPA in individuals with HTLV-1 infection. The hypothesis adopted was that when performing a task such as pointing to a visual target with their finger, individuals infected with HTLV-1 would have altered APA and CPA mechanisms. Our main findings confirmed the initial hypothesis, demonstrating that during the APA phase COP displacement was delayed, and presented a smaller amplitude in the infected group than in the control group. Moreover, the infected group presented with delayed activation of the ST muscle. During the CPA phase, the infected group required more time to reach the peak of COP displacement than the control group, which means that they took a longer time to stabilize their posture.

The pointing task has already been used in the literature [23,24,25], and it is known that the faster the movement is performed, the greater the magnitude of the induced disturbance, which influences the difficulty of the postural responses necessary to anticipate and recover postural stability [23,25]. Thus, the magnitude of the disturbance plays an important role in APA mechanisms, influencing their amplitude and duration [23,26]. Both groups presented similar peak velocities, demonstrating that they performed the entire task in a similar manner, that is, with the same perturbation magnitude. Therefore, the findings indicate that the determining factor for the significant differences between the groups was viral infection. 

When comparing the control and infected groups, it can be observed that both behaved similarly in relation to the COP trajectory, presenting a backward displacement before the beginning of the pointing movement and gradually returning to the anterior direction after its maximum posterior displacement. The COP results confirmed our hypothesis that the infected group had difficulties in preparing and recovering balance after the disturbance, which was demonstrated by a delayed COP_onset_, lower COP_APA_, and higher COP_timetopeak_ compared with the control group. Although the total displacement between the groups was similar, the control group was better prepared, anticipating COP_onset_ and covering most of the posterior COP displacement during the APA period (COP_APA_). Additionally, in the CPA phase, they recovered the initial position faster (COP_timetopeak_). 

More efficient anticipatory postural adjustments are associated with the time and ability to regain stability from the beginning to the end of the movement [27]. Considering this fact, the worse performance of the infected group can be attributed to changes in proprioceptive and postural control generated by the evolution of infection. Individuals infected with HTLV-1 are usually divided into asymptomatic and HAM/TSP, according to the diagnostic criteria of the WHO [5]; however, there are those who present incomplete signs of myelopathy, as classified by De Castro-Costa et al. (2006), and addressed by Koyama et al. (2019). Infected individuals, with or without symptoms, are susceptible to balance impairment [17], a functional ability that depends on motor and sensory components [6]. Proprioception is a sensory input for the regulation of postural stability [22] and is commonly altered in individuals with neurological symptoms, which can alter the accuracy of postural control in these individuals. The reduction in the quality of proprioceptive feedback can make it difficult to control the COM position, and cause ineffectiveness of the hip strategy. This strategy is complex, and depends on the interpretation of changes in hip and spine angles to calculate the COM position [28].

Changes in COP behavior were associated with delayed ST activation in the infected group. In addition, the infected group had a significantly lower percentage of trials with APAs (i.e., consistency of APA) in all muscles, except for RF. To our knowledge, there are no studies in the literature that have evaluated APAs in individuals with HTLV-1 through muscle activation; therefore, there are no findings regarding the muscle behavior of these individuals in a paradigm of balance disturbance. However, a delay in muscle onset has already been reported in individuals with lower back pain [29,30], multiple sclerosis [31], and post-stroke hemiparetics [32] when compared with control subjects. Delayed muscle activation has already been described as a dysfunctional joint strategy [33] and may be associated with challenging tasks [34]. Infected individuals have muscle delay in the paradigm used, and this may be associated with an anteroposterior imbalance that can generate insecurity and difficulty in maintaining balance.

Changes in APAs are related to poor balance control and maintenance, which can lead to a greater risk of falling, with the progression of postural control deficits.

This study has some limitations. The number of patients with HTLV-1 is low, due to the epidemiology of the disease. Moreover, from the 13 infected patients, five were not diagnosed as TSP/HAM, but only three of them had no neurological symptoms. This made it necessary to compare results between control (uninfected) and infected. It would be important to differentiate between the three categories (uninfected, HTLV-1, TSP/HAM) in future investigations. Despite this limitation, it was possible to observe significant findings that can help guide the investigation of balance control in individuals infected with HTLV-1, and physical therapy intervention based on exercises that stimulate better APA execution. Another limitation of the present paper is that no clinical history was available to find out about parameters such as drug abuse, for example. Further prospective studies with a larger population are needed to investigate the progression of these changes in APAs in individuals, considering the time of infection, viral load, onset of symptoms, and disease progression. In addition, it would be interesting to include a non-infected group with neurological alterations, for the purpose of comparing viral and non-viral impairment from a neurological point of view.

## 5. Conclusions

In conclusion, HTLV-1 infection can cause changes in APAs and CPAs. When performing rapid arm movements, the infected participants presented impaired APAs compared with the control group. Delayed lower-limb ST muscles and COP_onset_ resulted in this condition. In addition, their inability to achieve timely postural recovery (i.e., higher COP_timetopeak_) reflects delayed postural control during the compensatory phase. Hence, HTLV-1 infected individuals may face a higher risk of falling, in situations that require rapid recovery. Clinical assessment of this population should consider APAs.

## Figures and Tables

**Figure 1 viruses-14-02389-f001:**
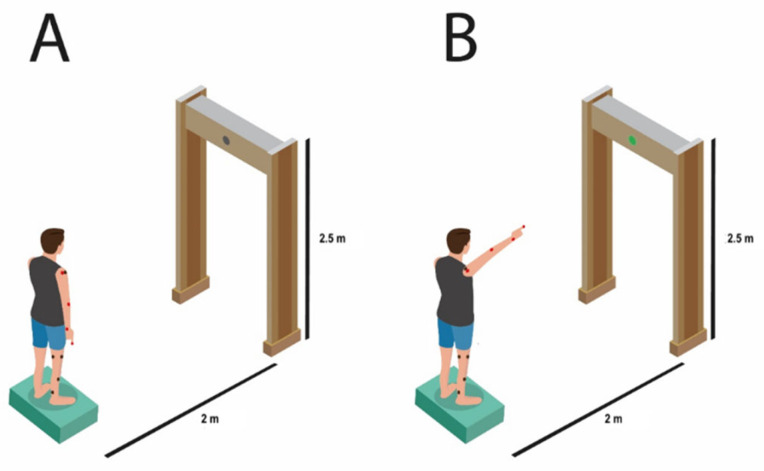
Demonstration of the experimental setup for the task. Figure (**A**) illustrates a participant in the initial position of the task, in a side view, with arms by the side of the body, and looking at the LED, which is off; Figure (**B**) represents the final position of the task, in which the individual stands with their arm raised and index finger pointing to the LED, which is on. LED: light emitting diode.

**Figure 2 viruses-14-02389-f002:**
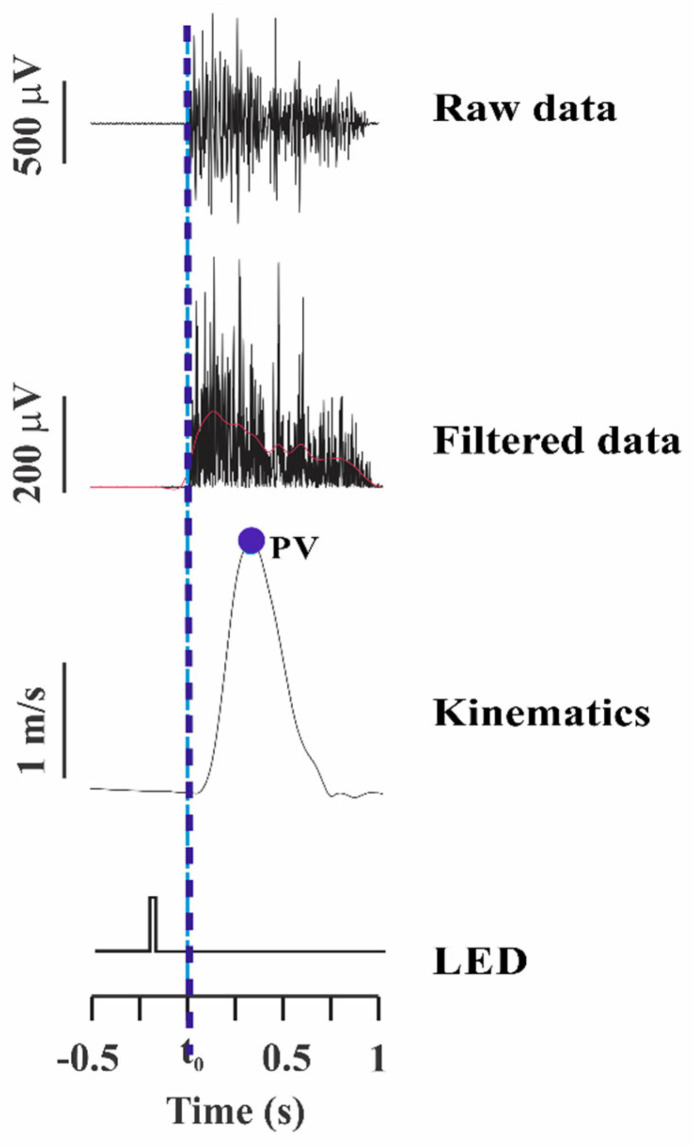
Illustration of deltoid muscle activation, raw and filtered data at 100 Hz low-pass. In addition, the marking of the velocity peak from the indicator trajectory is demonstrated, as well as the detection of the beginning of deltoid activation (T0, represented by the blue line), after stimulation performed with LED. LED: light emitting diode.

**Figure 3 viruses-14-02389-f003:**
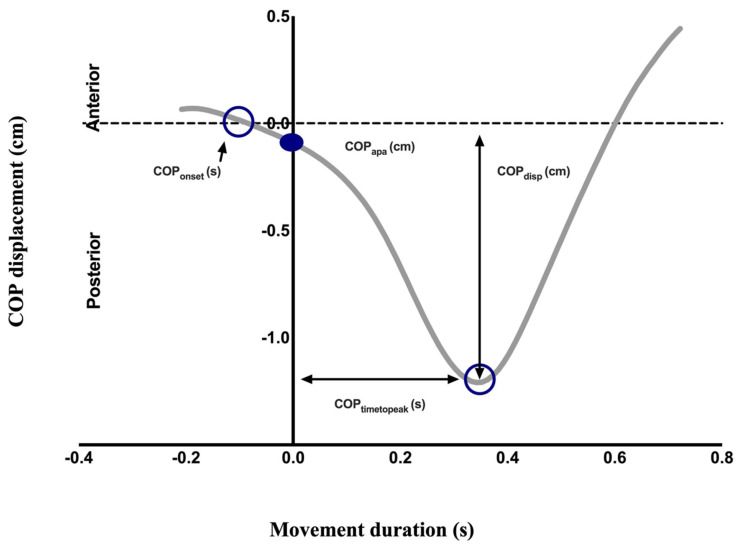
Anteroposterior displacement of the COP when the subject moves their arm. The dashed line represents the start of the movement. Four variables are demonstrated: (1) COP_APA_, posterior displacement amplitude of COP at T0; (2) COP_disp_, maximum backward shift after T0; (3) COP_onset_, time of onset of posterior displacement before T0; and (4) COP_timetopeak,_ time taken to reach maximum displacement. COP: center of pressure.

**Figure 4 viruses-14-02389-f004:**
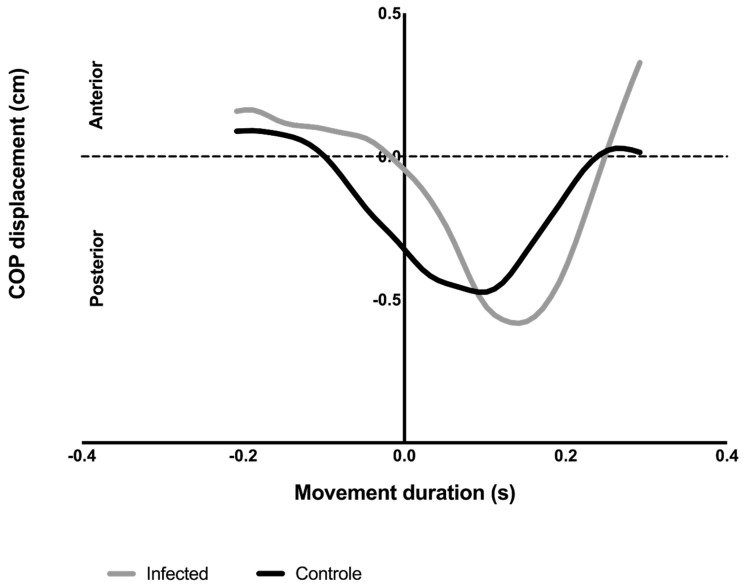
Representation of the behavior of COP_disp_ during arm movement of a typical subject of the control group and the infected group, during a single trial. Note that the control group participant had an earlier COP_onset_, a greater amplitude in COP_APA_, and took less time to reach maximum displacement after T0 (shortest COP_timetopeak_). No differences were observed between the COP_disp_ of the two groups. COP: center of pressure.

**Figure 5 viruses-14-02389-f005:**
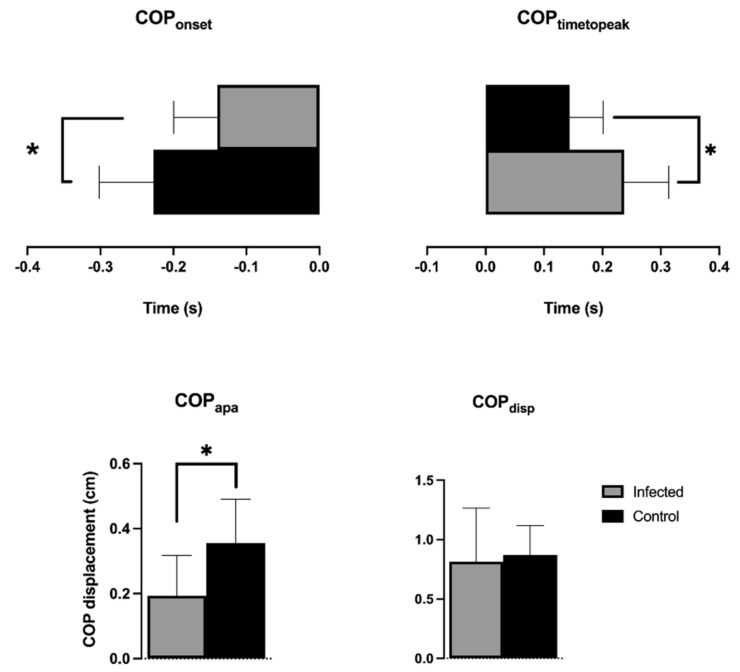
Statistical analysis of COP variables between the control and infected groups. The findings were considered significant when *p* < 0.05 (*). Data is presented as mean and standard deviation; the control group is represented in black, and the infected group in gray. COP: center of pressure.

**Figure 6 viruses-14-02389-f006:**
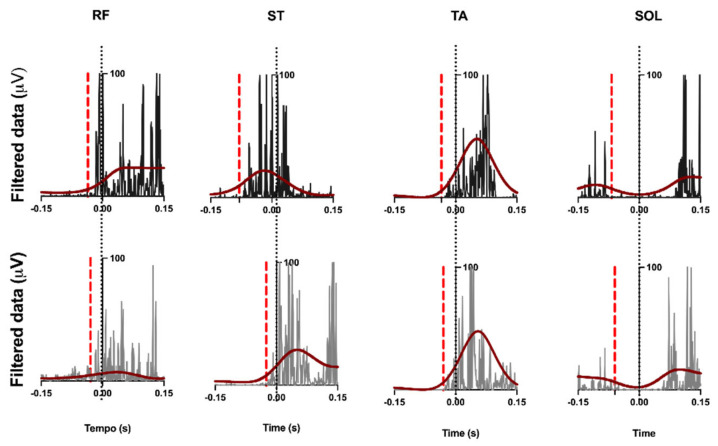
Data from control group (black) and infected group (gray). Rectified and filtered muscle activity at 6 Hz low pass of a typical participant in each group, recorded in a single trial. The solid vertical black line indicates the beginning of deltoid activity in the arm (T0), and the red dashed line indicates the beginning of activity of the evaluated muscles. Muscle abbreviations: ST, semitendinosus; RF, rectus femoris; SOL, soleus; TA, anterior tibial.

**Figure 7 viruses-14-02389-f007:**
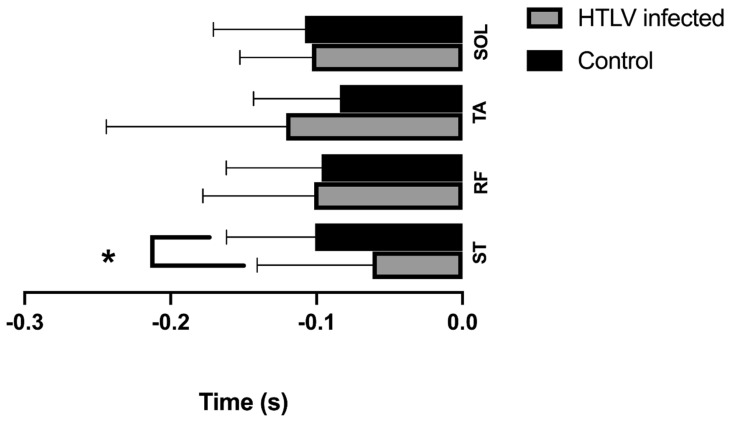
Statistical analysis of the EMG latency variable between the control and infected groups using the Mann–Whitney test. The findings were considered significant when *p* < 0.05 (*). Results show that only the ST muscle showed a significant difference between the groups, with a greater delay in the infected group. EMG, electromyographic; ST, semitendinosus.

**Table 1 viruses-14-02389-t001:** Characterization of the infected group in terms of diagnosis time and neurological symptoms.

Participant	Diagnosis	Diagnosis Time (Years)	Neurological Symptoms
1	HTLV-1	4	none
2	HTLV-1	3	bladder disorder
3	HTLV-1	5	none
4	HTLV-1	10	none
5	HTLV-1	7	bladder disorder sensory disorder backache
6	HTLV-1 TSP/HAM	12	spasticity in lower limbs sensory disorder sphincter disorders
7	HTLV-1 TSP/HAM	6	spasticity in lower limbs LL weakness sensory disorder sphincter disorders lumbosciatalgia
8	HTLV-1 TSP/HAM	12	spasticity in lower limbs LL weakness sensory disorder sphincter disorders erectile dysfunction lumbosciatalgia
9	HTLV-1 TSP/HAM	21	spasticity in lower limbs LL weakness sensory disorder sphincter disorders lumbosciatalgia
10	HTLV-1 TSP/HAM	9	spasticity in lower limbs LL weakness sensory disorder sphincter disorders erectile dysfunction lumbosciatalgia
11	HTLV-1 TSP/HAM	4	spasticity in lower limbs LL weakness sphincter disorders
12	HTLV-1 TSP/HAM	17	LL weakness sensory disorder sphincter disorders lumbosciatalgia
13	HTLV-1 TSP/HAM	8	LL weakness sensory disorder sphincter disorders lumbosciatalgia

HTLV, Human T-cell lymphotropic virus type 1; TSP/HAM, tropical spastic paraparesis/HTLV-1–associated myelopathy.

**Table 2 viruses-14-02389-t002:** Values, standard deviation and p-value of the variables: age, body mass index (BMI), sex, and peak velocity of the control and infected groups.

Variable	Control Group	Infected Group	*p*-Value
Age (Years)	48 (11.6)	50 (9.7)	0.18
BMI	25.65 (3.85)	26.40 (4.40)	0.11
Male/Female	5 M/8 F	4 M/9 F	0.34
Peak Speed (m/s)	5.649 (1.021)	5.169 (1.020)	0.30

**Table 3 viruses-14-02389-t003:** Statistical analysis of the percentage of EMG trials between the control and infected groups using the Mann–Whitney test. The ST, TA, and SOL muscles present significant differences between the groups, with a higher percentage density (%) of trials in the control group.

% Trials	Group Control	Infected Group	*p*
ST	60.923 (20.532)	20.786 (20.190)	0.0003 *
RF	30.846 (20.703)	10.929 (20.235)	0.0587
TA	50.923 (30.148)	20.929 (30.025)	0.0140 *
SOL	60.615 (20.694)	40.357 (20.437)	0.0194 *

*p* < 0.05 (*). ST, semitendinosus; RF, rectus femoris; SOL, soleus; TA, anterior tibial.

## Data Availability

The data that support the findings of this study are available from the corresponding author upon reasonable request.

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
