# Peer review of "Postural Adjustments in HTLV-1 Infected Patients during a Self-Initiated Perturbation"

_viruses, 2022, doi:10.3390/v14112389_

Round 1

Reviewer 1 Report

Almeida et al. evaluate the ‘Postural adjustments in HTLV-1 infected patients during a self-initiated perturbation’. Overall, it is a very interesting study, and my comments are following to be addressed :

1.      Do clinical history collected from the participants include parameter such as Drug abuse? If not, needs to be recorded. If yes, please mention in the text.

2.      When comparing results between control (Uninfected) and Infected, do infected refer to TSP/ HAM or all HTLV-1 infected individuals. It is important to differentiate between the three categories (Uninfected, HTLV-1, TSP/HAM) in the result/Discussion section rather than just two.

Author Response

Dear Editorial Board and reviewers,

The authors have received the evaluation and all considerations were taken into account. Thank you for helping us to improve the quality of the manuscript.

We´ve responded all the comments made, organizing the answer accordingly. Bellow, there is a list of itemized changes made, addressing each of the revision requirements,

All suggestions were attended and a new version of the manuscript is attached with the changes made (highlighted the within the text in yellow).

Reviewers' 1 comments:

Overall, it is a very interesting study, and my comments are following to be addressed :

  1. Do clinical history collected from the participants include parameter such as Drug abuse? If not, needs to be recorded. If yes, please mention in the text.

Authors. Thanks. Only neurological symptoms and demographic data were collected. we have recorded this information as a limitation in the new version

  1. When comparing results between control (Uninfected) and Infected, do infected refer to TSP/ HAM or all HTLV-1 infected individuals. It is important to differentiate between the three categories (Uninfected, HTLV-1, TSP/HAM) in the result/Discussion section rather than just two.

Authors. Thanks. You are correct and we tried to include this rationale as a limitation. The number of patients with HTLV-1 is low, due to the epidemiology of the disease. Moreover, from the 13 infected patients, 5 were not diagnosed as TSP/HAM, but only 3 of them had no neurological symptoms. This turned necessary to compare results between control (Uninfected) and Infected. But it would be important to differentiate between the three categories (Uninfected, HTLV-1, TSP/HAM) in future investigations

Reviewer 2 Report

The work was well conducted and has scientific merit.

The sample number of patients with HTLV is low, but it is expected, given the epidemiology of the disease.

The formation of analysis groups is not enough. The uninfected control group should be larger for a more robust comparison.

It would be interesting to include a non-infected group with neurological alterations for the purpose of comparing viral and non-viral impairment from a neurological point of view.

Author Response

Dear Editorial Board and reviewers,

The authors have received the evaluation and all considerations were taken into account. Thank you for helping us to improve the quality of the manuscript.

We´ve responded all the comments made, organizing the answer accordingly. Bellow, there is a list of itemized changes made, addressing each of the revision requirements,

All suggestions were attended and a new version of the manuscript is attached with the changes made (highlighted the within the text in yellow).

Reviewers' 2 comments:

The sample number of patients with HTLV is low, but it is expected, given the epidemiology of the disease. The formation of analysis groups is not enough. The uninfected control group should be larger for a more robust comparison.

Authors. Thanks. We have highlighted that in the new version. You are correct and we tried to include this rationale as a limitation. The number of patients with HTLV-1 is low, due to the epidemiology of the disease. Moreover, from the 13 infected patients, 5 were not diagnosed as TSP/HAM, but only 3 of them had no neurological symptoms. This turned necessary to compare results between control (Uninfected) and Infected. But it would be important to differentiate between the three categories (Uninfected, HTLV-1, TSP/HAM) in future investigations

It would be interesting to include a non-infected group with neurological alterations for the purpose of comparing viral and non-viral impairment from a neurological point of view.

Authors. Thanks. This is an interesting suggestion. We have added this as future directions.